# Evolution of Gestational Diabetes Mellitus across Continents in 21st Century

**DOI:** 10.3390/ijerph192315804

**Published:** 2022-11-28

**Authors:** Dominik Franciszek Dłuski, Monika Ruszała, Gracjan Rudziński, Kinga Pożarowska, Kinga Brzuszkiewicz, Bożena Leszczyńska-Gorzelak

**Affiliations:** 1Chair and Department of Obstetrics and Perinatology, Medical University of Lublin, 20-954 Lublin, Poland; 2Faculty of Medicine, Medical University of Lublin, 20-059 Lublin, Poland

**Keywords:** gestational diabetes mellitus, GDM prevalence, continents, GDM diagnosis, oral glucose tolerance test

## Abstract

Over the last few decades, several definitions of gestational diabetes mellitus (GDM) have been described. There is currently not enough research to show which way is the best to diagnose GDM. Opinions differ in terms of the optimal screening and diagnostic measures, in part due to the differences in the population risks, the cost-effectiveness considerations, and the lack of an evidence base to support large national screening programs. The basic method for identifying the disease is the measurement of glucose plasma levels which may be determined when fasting, two hours after a meal, or simply at any random time. The currently increasing incidence of diabetes in the whole population, the altering demographics and the presence of lifestyle changes still require better methods of screening for hyperglycemia, especially during pregnancy. The main aim of this review is to focus on the prevalence and modifications to the screening criteria for GDM across all continents in the 21st century. We would like to show the differences in the above issues and correlate them with the geographical situation. Looking at the history of diabetes, we are sure that more than one evolution in GDM diagnosis will occur, due to the development of medicine, appearance of modern technologies, and the dynamic continuation of research.

## 1. Introduction

In recent times a number of screening and diagnostic tests for gestational diabetes mellitus (GDM) have been used worldwide. There is currently not enough research to show which way is the best at diagnosing GDM. Opinions differ in terms of the optimal screening and diagnostic measures, in part due to the differences in population risks, cost-effectiveness considerations, and lack of an evidence base to support large national screening programs [1]. GDM usually manifests in the second half of pregnancy. Untreated glucose intolerance increases the risk of maternal and neonatal complications. Offspring whose mothers had GDM are more likely to be overweight, and obese, and they may develop glucose intolerance and type 2 diabetes mellitus (T2DM) in the future [2].

In ancient times, the practiced method of detecting diabetes mellitus was via an organoleptic urine assessment [3]. In the eleventh century, the basic method of identifying diabetes was still “uroscopy”, which consisted of the color, smell, and taste of the urine. The relationship between diabetes and metabolism was discovered at the beginning of the eighteenth century. At the turn of the eighteenth and nineteenth centuries, scientist Mathew Dobson combined the sweet taste of patients’ urine with excess sugar in the blood and urine [4]. 

The plasma glucose level may be determined when fasting, two hours after a meal, or simply at any random time. In addition, some tests involve drinking a glucose solution and measuring its concentration thereafter in the blood [5]. They are easy to administer and inexpensive. Currently, there are many scientific papers emphasizing the validity of screening for GDM in the first trimester of pregnancy [6,7,8,9]. To these tests a lot of attention is paid due to the fact of their repeatable effectiveness which can help in identifying patients at a high risk for GDM. The first-trimester glycated hemoglobin (HbA1c) assessment is pointed out to be an additional prognostic marker and should not be analyzed separately. Its insufficient sensitivity and specificity may not detect the whole population of patients [10,11]. 

The most common tests used to diagnose GDM are presented in Table 1. 

## 2. Australia and New Zealand

Australia and New Zealand are multiethnic territories with communities, who have a higher prevalence of incorrect BMI and GDM, such as Pacific women, or Māori women [13]. 

### 2.1. Australia

In 1991 the Australian Diabetes Society (ADS) recommended criteria to diagnose GDM: universal screening using a 50 g glucose challenge test (GCT) with a cut-off value ≥ 7.8 mmol/L (≥140 mg/dL); 75 g—oral glucose tolerance test (OGTT): fasting ≥ 5.5 mmol/L (≥99 mg/dL), and 2 h ≥ 8.0 mmol/L (≥144 mg/dL) [14]. This statement was ratified, accepted and released by the Australasian Diabetes in Pregnancy Society in 1998 (ADIPS98) [15]. The described screening method was used until 2014, when ADIPS released updated criteria according to the International Association of Diabetes and Pregnancy Study Groups (IADPSG). The new guidelines contain a universal OGGT at 24–28 weeks (fasting: ≥5.1 mmmol/L (≥92 mg/dL), 1 h: ≥10 mmol/L (≥180 mg/dL), 2 h: ≥8.5 mmol/L (≥153 mg/dL)) [16]. 

Researchers from Australia showed that after the adoption of the IADPSG standards the incidence of GDM increased from 20 to 75%, but it was also justified by improvements in the potential long-term benefits and perinatal morbidity [17,18,19]. Laurie et al. confirmed the almost quadrupled prevalence of GDM between 2010 and 2019 in Australia. A total of 14,225 GDM cases were diagnosed in 2010 and 40,848 cases in 2019 (one fifth were cases of repeated diagnoses of GDM) [20], which resulted in approximately 14% incidence of GDM in Australia [21]. The scientists believed that a few things contributed to this situation: changes in the criteria of a GDM diagnosis, rising rates of obesity and overweight, diversity of ethnicity, and increasing maternal age [20].

### 2.2. New Zealand

In 1998, the New Zealand Society for the Study of Diabetes (NZSSD) adopted diagnostic criteria, with values that were higher than those in Australia: 50 g—GCT with cut-off value ≥ 7.8 mmol/L (≥140 mg/dL) at 24–28 weeks; 75 g—OGTT, fasting ≥ 5.5 mmol/L (≥99 mg/dL) and 2 h ≥ 9.0 mmol/L (≥162 mg/dL) [15]. In December 2014, the New Zealand Ministry of Health (NZMOH) recommended multi-layered screening guidelines. According to them, every pregnant patient at <20 weeks of gestation should be offered a glycated hemoglobin (HbA1c) assessment to identify undiagnosed pre-existing diabetes mellitus. If the HbA1c is ≥6.7%, a pregnant woman should be treated as having probable diabetes mellitus in pregnancy (DIP). When a patient has an HbA1c ≤ 5.8% at 24–28 weeks of gestation, a 50 g GCT should be offered. When an HbA1c is 5.9–6.6%, a 75 g OGGT is performed [22]. 

The prevalence of GDM in New Zealand has not been definitively understood, or reported according to small studies performed in small catchment areas [23,24,25,26,27], but the latest studies assessed it at approximately 6% (5.7–6.2%). This score was confirmed in different ways [28,29]. Chepulis et al. presented that GDM more likely affects women at an advanced age, as well as Māori, Pacific, and Asian women [28].

## 3. Africa

### 3.1. Attempts at Assessing the Prevalence, Screening Methods and Risk Factors of GDM throughout the Entire African Continent

On the African continent, a few ways diagnosing GDM diagnosis have still exist. The WHO 1985, WHO 1999, WHO 2006, CC criteria, ADA 2003/2004, WHO/IADPSG 2013 or individual protocols are taken into account [30,31]. This lack of uniformity in GDM diagnosis protocols between countries and within countries create problems when comparing the prevalence of GDM on this continent. Even after changing the criteria for a GDM diagnosis in the WHO/IADPSG 2013, the overall prevalence of GDM is still unknown. Several studies concerning GDM in Africa were performed to try to fill in the gap regarding GDM prevalence, risk factors, outcomes and management, but the majority of them focused on specific populations [30,31,32]. One thing was obvious: for all of the studies, the GDM prevalence was higher, which was confirmed by Olumodeji et al. [33]. This is connected with urbanization, changing lifestyles, and newer and more common GDM screening methods [30,31]. 

Mwanri et al., in their review, suggested that the incidence of GDM in sub-Saharan Africa is approximately 14% among high- risk patients [32], but the prevalence of GDM in the general population throughout all African countries was like a blank space. According to Natamba et al.’s systematic review, the incidence of GDM was assessed at 3% before 2010 and approximately 13% between 2010 and 2018, but if the IADPSG criteria were used the prevalence was approximately 16% in sub-Saharan Africa [30]. 

Muche et al. were the first researchers, who tried to present data from all parts of Africa. According to them, the pooled prevalence of GDM in Africa was 13.61% (OR: 95%CI: 10.99–16.23; I^2^ = 96.1%) and 14.28% (95% CI: 11.39–17.16; I^2^ = 96.4%) in subSaharan countries. The lowest prevalence was in Northern Africa at 7.57% (95% CI: 5.89–9.25) and the highest was in Central Africa at 20.4% (95% CI: 1.55–38.54) [34]. 

The risk factors for GDM are not well documented in African countries, but typical risk factors for GDM should be the same as in other populations [32,35]. Additionally, some local drivers such as infections or undernutrition may play a role in the higher risk for GDM development [36]. According to Natamba et al. the most important risk factors for GDM in subSaharan Africa were: GDM, stillbirth, abortion, macrosomia in previous pregnancies, family history of T2DM and hypertension. Additionally, an age greater than 25 years, overweight, or obesity, and mutliparity also increased the risk of GDM, but primiparity decreased this risk. Other parameters (e.g., HIV infection) were not statistically significant [30]. 

According to Muche et al., overweight/obesity (OR = 3.51; 95% CI: 1.92–6.40), family history of diabetes mellitus (OR = 2.69; 95% CI: 1.84–3.91), macrosomia (OR = 2.23; 95% CI: 1.84–3.91), stillbirth (OR = 2.92; 95% CI: 1.23–6.93), abortion (OR = 2.21; 95% CI: 1.68–2.92) GDM in previous pregnancy (OR = 14.16; 95% CI: 2.39–84.48), and hypertension (OR = 2.49; 95% CI: 1.35–4.59) were positively correlated with GDM [34].

### 3.2. The Prevalence, Screening Methods and Risk Factors of GDM in Selected Countries or Parts of Africa

Egbe et al., using WHO/AIDPSG 2013, confirmed in their study that the highest prevalence of GDM is in Central Africa-Cameroon (20.4%). These results were connected with an epidemic of obesity caused by incorrect dietary habits and a low level of physical activity. Furthermore, they agreed with other researchers that macrosomia (OR = 8.5, 95% CI: 3.8–19.0, *p* < 0.001), past history of unexplained stillbirth (OR = 5.7, 95% CI: 2.5–12.9, *p* < 0.001), BMI ≥ 30 kg/m2 (OR = 6.2, 95% CI: 2.9–13.1, *p* < 0.001), and advanced maternal age (OR = 3.4, 95% CI: 1.7–7.0, *p* < 0.001) were significantly related to GDM [37].

Mghanga et al. presented data from southern Tanzania. They used WHO/IADPSG 2013 criteria to diagnose GDM in their cross-sectional study. The GDM prevalence was lower (4.3%) than the pooled prevalence in all of Africa. They also found significant correlations between overweight/obesity (*p* < 0.001), past history of preterm delivery (*p* < 0.001), macrosomia (*p* < 0.001), stillbirth (*p* < 0.001), alcohol consumption (*p* < 0.001), and family history of DM (*p* < 0.001) and GDM [38]. 

Nwali et al. analyzed which type of GDM screening was better- selective or universal among 400 patients from Nigeria. The selective screening was defined as 75 g OGTT between 24 and 28 weeks of gestation, performed in a patient with ≥1 risk factors for GDM. The results showed that the GDM incidence was higher with the universal screening (11.51%) than for the selective screening (7.93%), which provided a miss rate of 31.11%. They also found positive correlations between hyperglycemia and previous history of GDM and hypertension, weight ≥ 90 kg, and age ≥ 35 years [39]. 

Al-Rifai et al. in their article presented data from North Africa and the Middle East (MENA region) between 2000–and 2019. The weighted incidence of GDM in North Africa was 13.5% (95% CI: 7.4–20.9, I^2^ = 98.9%). This prevalence was 32% higher in North African countries with a maternal mortality ratio (MMR) > 100/100,000 live births than in North African countries with an MMR ≤ 100/100,000 live births [40]. 

On the other hand, the newest publication from Gabon presented a comparison between two diagnostic guidelines for GDM: WHO 1999 and WHO/IADPSG 2013, and a disagreement with the thesis described in almost all articles. They showed that the GDM prevalence was higher using the WHO 1999 (10.4%) than using the WHO/IADPSG 2013 criteria (4.5%). The researchers also found an association between GDM and overweight/obesity (*p* = 0.0498), GDM in previous pregnancy (*p* < 0.0001), and frequency of C-section (*p* = 0.0334) and macrosomia (*p* = 0.0082). Additionally, their data showed that advanced maternal age was connected with GDM only with high parity [41].

## 4. North America

According to the International Diabetes Federation (IDF) for 2021, the pooled prevalence of GDM in North America and the Caribbean (NAC) was assessed as 7.1%. The result was the lowest among the IDF regions. This meta-analysis used data from the United States of America, Mexico, Canada, and Barbados [42].

### 4.1. The United States of America

According to the American Diabetes Association (ADA), two strategies can be used to diagnose GDM:

The one-step 75 g OGTT using IADPSG/WHO criteria. 

The two-step method with a no fasting screen of 50 g, followed by a 100 g OGTT for screened positive patients [43] 

These both methods are characterized by various advantages and disadvantages. A recent comparative study analyzed these two screening strategies. The results showed that using the one-step strategy, GDM was diagnosed in doubled number of women in comparison with the two-step method. Nevertheless, there was no disparity between maternal and perinatal outcomes, while more pregnant women were treated for GDM after diagnosis with the one-step method [44]. This increase in the number of GDM cases occurred, due to the fact that only one elevated value is required to make the diagnosis [45].

In spite of the ADA endorsement for one-step OGGT, the National Institutes of Health (NIH) in the US recommend the two-step criteria for a GDM diagnosis [46]. These criteria were adopted by the American College of Obstetricians and Gynecologists (ACOG). During the decision-making process, the main factors were a lack of clinical evidence of the impact of the one-step strategy on pregnancy outcomes and the possibility of negative results when diagnosing a great number of patients with GDM using the one-step strategy. The consequences would be increased health care costs and utilization of medicalization [47]. Moreover, using a 50 g GCT is easier to perform due to the lack of requirement of fasting [48].

In connection with the increased frequency of maternal obesity, an increase in the occurrence of GDM has been observed as well. Maternal obesity is a well-known, independent risk factor for GDM. While in 1961 the incidence of GDM was estimated to be less than 1% [49,50], it increased from 0.3% in 1979–1980 to 5.8% in 2008–2010 [51]. Nowadays, the prevalence of GDM was estimated by the Centers for Disease Control (CDC) to be approximately 10% in the United States (US), using the two-step OGTT as a screening. In turn, using the one-step 2-h 75 g OGTT approximately 17.8% of pregnant women would be classified as having GDM, which would almost double the frequency of GDM in the US [4].

Interesting data were presented by Tsai et al. They analyzed 4735 live births in Hawaii between 2009–and 2011 with a division by ethnicity. The pooled GDM prevalence was 10.9%, but a higher incidence of GDM was confirmed in Filipina (13.1%) and Pacific Islander/Hawaiian (12.1%) pregnant women. The lowest rate of GDM was among White women (7.4%). Filipina, Pacific Islander/Hawaiian, and other Asian pregnant patients had increased risk of GDM versus white women using a bivariate analysis [52].

### 4.2. Mexico

Nowadays, in Mexico, there is no protocol, recommended to perform GDM screening. The National Center for Health Technology Excellence (CENETEC) only presents the possibilities for diagnosing GDM [53] (Figure 1).

According to Dainelli et al. the most popular screening method is the 75 g one- step OGTT (46.6% of the total cases) [54].

In Mexico, no official data on the GDM prevalence are available. However, in 1988, the first paper was published reporting the incidence of GDM—the level estimated was at 4% [55]. A growing trend of GDM prevalence has been observed [56,57,58,59,60,61]. The incidence achieved more than 30% in 2016 [62]. The data were collected in 4 major cities in Mexico: (Mexico City, Guadalajara, Monterrey, and Merida) in 2017, and an overall predicted GDM prevalence at the level of 23.7% was reported [54].

### 4.3. Canada

GDM is the most frequent endocrinopathy during pregnancy in Canada [63], but for many years, there has been no consensus regarding its diagnosis. The guidelines presented by the Canadian Diabetes Association (DC) and the Society of Obstetricians and Gynaecologists of Canada (SOGC) had differences regarding a few things. Over the last 30 years, they have been changed by the SOGS 4 times (1992, 2002, 2016, and 2019) [64,65,66,67] and by the DC 5 times (1998, 2003, 2008, 2013, and 2018) [68,69,70,71,72]. Just after the last modifications, some type of agreement appeared, but two methods are still used to identify GDM: a two-step strategy and a one-step strategy. Both organizations prefer the two-step strategy with a 75 g—OGTT and an abnormal value to diagnose GDM: fasting glucose: ≥5.3 mmol/L (≥95 mg/dL), 1 h: ≥10.6 mmol/L (≥191 mg/dL) and 2 h: ≥9.0 mmol/L (≥162 mg/dL). The one-step approach is an alternative method [67,72]. Analyzing the data, the incidence of GDM changed from 3.8 to 6.5% (SOGC) or 2.0–4.0% (DA) in 1992 to 7.0% using the two-step method or to 16.1% using the one-step method. This drastic rise in GDM prevalence by four-fold in the number of pregnant women diagnosed with GDM, occurred over the last 20 years [64,65,66,67,68,69,70,71,72]. Additionally, it cannot be forgotten the increased number of obese, ethnically diverse, and maternal aged patients, who had a large contribution to the increase in the GDM prevalence [73]. These data were confirmed in large population-based studies performed in Ontario [74], Saskatchewan [75], and Alberta and British Columbia [76].

Interesting results were presented by Poirier et al. in an article from northwestern Ontario. They performed an 8-year retrospective analysis, that found that the average annual GDM prevalence was 12% (8–17%), two times higher than in all of Ontario. In 90% of the cases, they used a two-step screening protocol to diagnose GDM [77].

## 5. South America

The problem that exists in South America is a lack of uniform criteria for diagnosing GDM and a lack of population-based incidence studies on the condition. Some Latin American countries have adopted the IADSPG criteria, but many of them have made locally adapted modifications due to the practical problems of implementing them in a resource-constrained environment [78].

According to the IDF Diabetes Atlas 2021 the standardized pooled prevalence of GDM in South America and Central America (SACA) was 10.4%. These data were calculated after analyzing 4 studies from this region (i.e., Argentina, Brazil, Chile and Cuba) [42]. Similar results for GDM prevalence at 10.3% were presented by Leonco et al. in their prospective study performed in French Guiana, an overseas department of France, in South America [79].

### 5.1. Brazil

The Brazilian Diabetes Society (SBD) recommend the guidelines of the (IADPSG). According to the recommendations of the Brazilian Ministry of Health, at least one of the following criteria must be met to diagnose GDM in the OGTT test: glycemia (fasting) ≥ 5.1 mmol/L (≥92 mg/dL) and ≤6.9 mmol/L (≤125 mg/dL); glycemia 1 h after overload ≥ 10 mmol/L (≥180 mg/dL); glycemia 2 h after overload ≥ 8.5 mmol/L (≥153 mg/dL) and, <11.1 mmol/L (<200) mg/dL. In a study performed among users of the Brazilian Unified Health System (UHS) in the city of Caxias do Sul, State of Rio Grande do Sul, the estimated prevalence of GDM was 5.4%. Additionally, pregnant women who were obese, ≥35 years, and those with three or more pregnancies were more likely to develop GDM during pregnancy than women with a normal BMI, <35 years, and primiparous [80].

A retrospective cohort study was conducted by Sampaio et al. using the medical records of women diagnosed with dysglycemia during pregnancy between January 2015 and July 2017 at the Specialized Outpatient Clinic for Pregnancy Endocrinopathies at Taguatinga Regional Hospital, Federal District at a public health center in Brazil. Approximately 70% of the patients were overweight or obese, 78.6% of them had GDM and 21.4% had diabetes mellitus in pregnancy [81].

### 5.2. Argentina

The criteria for the diagnosis of GDM were established by the Latin American Diabetes Association (ALAD) in 2007 and are still used nowadays. According to them, during the first visit, pregnant patients have a fasting glycemia measurement. If the result is ≥100 mg/dL, the test will be repeated for the next 3 days. GDM can be diagnosed, when two measurements are ≥5.5 mmol/L (≥100 mg/dL). If not, a 75-g OGTT is recommended between 24–and 28 weeks of gestation. If a patient has glycemia < 7.77 mmol/L (<140 mg/dL) in the second hour, she will be considered as a woman without GDM, while if the value is ≥7.7 (≥140 mg/dL), GDM is diagnosed. If a patient has a correct 75-g OGTT, but has GDM risk factors, the OGTT will be repeated between 31–and 33 weeks of gestation [82].

A clinical study, published in 2020, compared the prevalence of GDM in Argentina according to the ALAD and IADPSG criteria. The results showed that the GDM incidence in Argentina using the IADPSG criteria was 2.54 higher (24.9%) than the prevalence according to the ALAD guidelines (9.8%). No differences were found in BMI, maternal age or pregnancy length between pregnant patients diagnosed using the ALAD and IADPSG guidelines. Additionally, the authors presented an almost two-fold increase in the GDM prevalence [83] in comparison to the years: 1995 (5.0%) [84] and 2009 (5.8%) [85].

### 5.3. Chile

The perinatal care guidelines, published in 2015 by the Ministry of Health in Chile, established that pregnant women with GDM are diagnosed with a 75 g OGTT at 24–28 weeks of gestation; fasting glucose values ≥ 5.55 mmol/L (≥100 mg/dL) and/or ≥7.77 mmol/L (≥140 mg/dL) 2 h after OGTT indicate GDM. Additionally, fasting glucose values between 5.55 (100 mg/dL) and 6.94 mmol/L (125 mg/dL) in the first trimester of pregnancy mandate a diagnosis of GDM. Before this change only the OGTT was recommended to diagnose GDM [86]. The biggest study analyzing GDM prevalence in Chile was performed between 2002–2015, 86,362 pregnant women were included. The mean prevalence of GDM was 7.6% (95% CI: 7.5–7.8) with increasing values from 4.4% in 2002 to 13.0% in 2015. Additionally, researchers presented the risk factors for GDM: age, civil status, education, family history of T2DM, personal history of GDM, preeclampsia, hypertension, pre-gestational nutritional status, smoking, and alcohol [87]. Another study estimated the GDM prevalence in Chile at 6.6% among pregnant women [88].

### 5.4. Ecuador

In Ecuador, an interview is used to identify pregnant women with intermediate (overweight) and high-risk factors (i.e., obesity, polycystic ovarian syndrome, family history of diabetes, and macrosomia in previous pregnancies). It works as a screening test, then a fasting glucose level is ordered for patients with a positive interview in the first trimester of pregnancy. It is interpreted as follows: greater than 7 mmol/L (126 mg/dL): pre-existing diabetes, between 5.1 and 7 mmol/L (92 and 126 mg/dL): GDM; normal result. If the result is in the norm, the 75 g—OGTT is performed between 24–and 28 weeks of gestation. If the result is abnormal, a 75 g—OGTT is suggested to be conducted immediately. The GDM prevalence in Ecuador is estimated at approximately 10% [89].

### 5.5. Colombia

Colombian recommendations say that a pregnant woman has her first prenatal visit between 7 and 12 weeks of gestation, during which first screening—fasting or casual of their blood glucose should be performed to assess the situation and prepare for future management. If the fasting glucose level is ≥7 mmol/L (≥126 mg/dL) pre-gestational diabetes mellitus (PGDM) is diagnosed, when it is between 5.3–and 6.94 mmol/L (92–125 mg/dL) GDM is diagnosed. Patients with a fasting glucose level of <5.3 mmol (92 mg/dL) will have a 75 g OGGT between 24–and 28 weeks of gestation; if casual glucose level ≥ 200 mg/dL PGDM is diagnosed, or when <200 mg/dL—a 75 g—OGTT between 24–and 28 weeks of gestation is recommended [90].

In a study performed between 2013 and 2017 in Zapatoka, the Santander mixed criteria were used to diagnose GDM, and the prevalence was estimated at 4.46% [91]. In another study from Colombia, using the IADPSG/WHO 2013 criteria, which took place between July 2017 and March 2018 at the Hospital Universitario San José in Popayán, the estimated prevalence of GDM was 16.32%. This result was similar to most studies that used the IADPSG criteria, but it was higher than the Colombian results described previously. Additionally, they analyzed the GDM risk factors. The data showed that: age > 35 years, indigenous race, history of fetal macrosomia, and BMI > 25, had a significant association with GDM [92].

### 5.6. Peru

There are poor data on the GDM prevalence in Peru [93], despite the fact, that a large prospective study, assessing the prevalence and risk factors of GDM using the IADPSG/WHO criteria was performed. The results presented a 16% GDM prevalence among pregnant women. Additionally, researchers found that mid-pregnancy obesity, and family history of diabetes, had increased odds ratios of GDM [94].

### 5.7. Guyana

Guyana is a former British colony, that has a mutliethnic population with a domination of 40% Indo-Guyanese and 30% Afro-Guyanese, which are at a high risk for diabetes mellitus. According to a national study, the prevalence of diabetes mellitus was 14.9%. Lowe at al. presented data from a program based on IDF Women in India with GDM Strategy (WINGS) using the 75 g—OGTT. Three research centers took part in it: Georgetown Public Hospital Corporation (GPHC) and two associated heath centers in the (HCs). The GDM prevalence in Guyana, in GPHC, and in HCs was 22.1%, 25.9%, and 2.6% respectively [95].

### 5.8. Uruguay

A cross-sectional study from Uruguay analyzed data from 42,663 pregnant women, who delivered in 2012. Perinatal Information System (SIP) records were used. The GDM prevalence was estimated at 22% using only fasting glucose level ≥ 5.3 mmol/L (≥92 mg/dL). The glucose levels after 1 h, and 2 h from the OGTT were not available in the SIP [96].

### 5.9. Trinidad and Tobago

The national prevalence in Trinidad and Tobago was estimated at 14.5% using the IADPSG/WHO 2013 criteria [97]. In a retrospective observational study conducted by Clapperton et al. a marked increase in GDM prevalence was observed. The GDM frequency in 2005, 2006, and 2007 was 1.67%, 4.58%, and 6.67% respectively with a mean incidence of 4.31%, and a predicted value of 9.31% for 2008. In addition, they found that age, ethnicity, family history, GDM in previous pregnancies, and obesity were risk factors for GDM [98]. In contrast, a cross-sectional study from north central Trinidad, which lasted from January 2012 to December 2016 estimated the prevalence of GDM at 2%, but the authors mentioned that the real prevalence might be higher and further investigations were needed [97].

### 5.10. Paraguay, Suriname, Bolivia, Venezuela

Studies on the GDM prevalence and screening methods in Paraguay, Suriname, Bolivia, and Venezuela were not found. They probably adopted the WHO/IADPSG criteria with regional modifications to diagnose GDM [78,96].

## 6. Europe

Seventeen studies were analyzed by the IDF to assess the regional standardized GDM prevalence in Europe—7.8% [42]. According to a meta-analysis prepared by Paulo et al., the overall weighted prevalence of GDM from 24 European countries was 10.9%. The highest GDM prevalence was in Eastern Europe—31.5%, and the lowest was in Northern Europe—8.9%. The values for Western Europe and Southern Europe were 10.7% and 12.3%, respectively. Women age > 30 years, overweight/obesity, and a GDM diagnosis in the third trimester were risk factors for the increased prevalence of GDM [99].

### 6.1. Poland

In Poland every pregnant woman has a fasting glucose test on her first prenatal visit. If a patient has risk factors for GDM, or an incorrect fasting glucose value, the OGTT is recommended. If their fasting glucose level or OGTT is at norm, the patient will have an OGTT performed according to the standard screening between 24–28 week of gestation. This scheme was prepared according to the IADPSG/WHO 2013 [100]. Bomba-Opoń et al. in a retrospective multicenter cohort study presented the most actual data regarding the GDM prevalence in Poland—at 6.62% and it increased two-fold over the last 20 years [100]. Similar results were described by Wojtyla et al. In their study, the GDM incidence in 2012 was 4.0% and in 2017 it was 6.2% [101]. A study performed by Cichocka and Gumprecht found that more pregnant women were diagnosed with GDM based on the fasting glucose level [102].

### 6.2. Spain

In Spain, the criteria of the National Diabetes Data Group (NDDG) from 1979 are still used. According to them, in all pregnant women between 24 and 28 weeks of gestation, and in patients with GDM risk factors during the first trimester of pregnancy, a screening test is performed with a 50 g glucose. Women with a positive screening test with a 1-h blood glucose > 140 mg/dL (7.7 mmol/L) underwent a confirmatory 3-h, 100 g—OGTT. GDM is diagnosed with two abnormally high values of the following thresholds: fasting glucose level ≥ 105 mg/dL (5.8 mmol/L); 1-h, ≥190 mg/dL (10.5 mmol/L); 2-h, ≥165 mg/dL (9.1 mmol/L); 3-h, ≥145 mg/dL (8.0 mmol/L) [103,104].

In a study comparing the criteria from 1979 and the IADPSG/WHO 2013 guidelines, the results were surprising: the GDM prevalence changed dramatically from 10.6% using old criteria to 35.5% using the new criteria [105]. López-de-Andrés et al. in a population-based study performed between 2009 and 2015 using the DDG criteria assessed GDM at 5.27% [106]. These results are similar to data presented by Gortazar from Katalonia—the mean GDM prevalence between 2009 and 2015 was estimated at 4.8% [107]. Additionally, an increasing trend in the GDM frequency from 3.81% in 2009 to 6.53% in 2015 was observed [107]. The study performed by Melero et al. using the IADPSG criteria, analyzed the nutritional intervention, the Mediterranean diet and its influence on GDM, the results showed lower GDM prevalence in the Mediterranean diet group, than in the control group at 14.8% and 25.8%, respectively [108].

### 6.3. Portugal

The Portuguese Gynecological Society uses the IADPSG criteria for screening and diagnosing GDM [109]. GDM prevalence was estimated at 3.4% in 2005, 6.7% in 2014, and 8.8% in 2018, which is 2.58-fold increase in frequency after changing the diagnostic guidelines for GDM. The highest prevalence was estimated among women with GDM ≥ 40 years [110,111].

### 6.4. France

France accepted new IADPSG criteria in 2010 and uses one-step strategy to screen for GDM. It has had an influence on the GDM prevalence, which was estimated at approximately 5% in 2004–2005 [112], 11.6% in 2013 [113], and even up to 22.5% in 2016 [114]. Miailhe et al. compared selective and universal GDM screening using the IADPSG criteria. The results showed that one-sixth of GDM cases would have been missed by selective screening [115]. A study from Brest confirmed that obesity is related to GDM (OR 5.83, 95%CI: 4.37–7.79) [116].

### 6.5. Italy

In 2010 Italy accepted and adopted the IADPSG/WHO 2013 as the universal screening criteria for GDM, but in September 2011, the Italian Public Health Authority changed the GDM guidelines and decided on selective screening. The reason was due to the fact that the level of GDM diagnoses was too high using the new criteria. Women, who were classified as healthy in previous classification, became GDM affected by new screening [117,118]. Lacaria et al. checked the application and effectiveness of GDM selective screening. Their results showed that the prevalence of GDM was 10.9% and 25% higher than when using the old criteria. Additionally, most of the tests were performed between 24 and 28 weeks of gestation, which departs from the new Italian guidelines (between 16 and 18 weeks of gestation in GDM high-risk pregnancies, between 24 and 28 weeks of gestation in medium-risk pregnancies; no OGTT in low-risk pregnancies) [118].

Di Canni et al. confirmed the GDM prevalence rate—at 11% when analyzing data from Tuscany. Additionally, they consented that with the lack of proper screening—only 55% performed OGTTs according national guidelines. Moreover, the researchers also suggested the need for universal screening, due to the relatively high (7.0%) GDM prevalence level among non-eligible pregnant patients [117].

Zanardo et al. compared two groups: mothers giving birth before the COVID-19 pandemic and during the pandemic. The results showed a statistically significant increase in the GDM prevalence rate from 9.0% in 2019 to 13.5% in 2020 [119].

### 6.6. Germany

In 2012 two-steps screening of GDM with a 50 g—GCT and a 75 g—OGTT was inaugurated in Germany [120]. Until this time, different strategies for GDM diagnosis existed; therefore, the prevalence rates were very variable between 2% and approximately 18% [121,122,123]. Melchior et al. analyzed data from 2014 to 2015. The overall prevalence was estimated at 13.2%, with the highest rate among women ≥ 45 years at 26% [120].

Before starting universal screening in Germany, a trend of increasing GDM prevalence rates was observed [124]. Researchers from the North Rhine region analyzed data from 12 months before and 12 months after the introduction of universal screening with an incidence of 6.02% before, and 6.81% after. They found that the prevalence rate relatively increased by approximately 13.12% after the beginning of universal screening. Additionally, the GDM prevalence was the highest among women between 36 and 40 years [125]. Reitzle et al. [126] presented data on the GDM incidence from 2016 to 2018, in 2016 was 5.3% and in 2018 was 6.8%. Researchers signaled two things: the data were taken from hospital records, and several percent of German women still have not had a GDM screening [126].

### 6.7. Greece

The National Health Guidelines in Greece recommend the IADPSG/WHO 2013 criteria to diagnose GDM. The GDM prevalence, as assessed by Varela et al., was 14.5% [127]. A similar result was estimated by Papachatzopoulou et al. in their prospective cohort study, at 11.5% [128]. Vasileiou et al. showed that a GDM diagnosis is related to seasons, the highest prevalence was assessed in the summer, and the lowest in the winter [129]. Other researchers from Greece suggested that there is higher rate of GDM prevalence among in vitro fertilization pregnancies [130].

### 6.8. Switzerland

Switzerland also changed the GDM diagnosis criteria and adopted the IADPSG guidelines in 2010. Before this date, the two-step criteria were used to diagnose GDM. Orecchio et al. in 2004–2005 performed a study among 1042 pregnant women, and the prevalence was 4.8%. They found a statistically significant association between GDM and Asiatic origin, and GDM diagnosed during a previous pregnancy [131]. Huhn et al., in their cohort study compared two groups of pregnant patients: before (period 1) and after (period 2) changing the criteria. The results showed an enormous increase in the GDM prevalence rate from, 3.3% in period 1 to 11.8% in period 2 [132]. A similar study was performed by Aubry et al., in which GDM incidence rates were assessed for two periods of time, between 2005 and 2010, and between 2012 and 2017. The threefold increase in GDM prevalence was observed between these two periods from 2.7% in the first period to 8.3% in the second period [133].

### 6.9. Austria

Austria adopted the IADSPG criteria in 2010. The OGTT is administered between 25 and 28 weeks of gestation [134]. Researchers from all parts of the country observed that the GDM prevalence rates have changed. Before the introduction of the new guidelines approximately 7–10% pregnant women were affected by GDM according to the literature [135]. In a study performed by researchers from Linz, GDM prevalence was assessed at 16.8% using the new criteria for GDM diagnosis [134]. Similar results were presented by Kotzaeridi et al., with an incidence of GDM at 17.8% [136]. On the contrary, the results presented by Muin et al. were much lower. In 2008–2010, the estimated GDM prevalence was 2.9%; in 2011–2019 it was 4.38% [137].

Other group of researchers from Austria also analyzed the GDM prevalence in triplet pregnancies. The rate was absolutely higher (31.7%) than in singleton pregnancies [138].

### 6.10. Czech Republic/Czechia

In the Czech Republic/Czechia, there has been a discussion regarding the IADPSG criteria. During 2014 and 2015 Czech medical societies gradually adopted this criteria. Before changing the criteria Czech doctors alarmed that the GDM prevalence would be higher using the new guidelines [139]. As an example, a study performed by Anderlová et al. in 2014 can be presented. The Czech criteria with cut-off values for plasma glucose levels: for fasting at—5.6 mmol/L (100 mg/dL), 1 h—at 8.9 mmol/L (160 mg/dL), and 2 h—at 7.7 mmol/L (140 mg/dL) provided the GDM prevalence of 22.26%. The IADPSG guidelines resulted in a 31.89% incidence of GDM [139]. In opposition to these results, Krejčí et al. found that using the old Czech criteria, the GDM prevalence was 20.3%, and according to the new guidelines the incidence was 14.3% [140].

### 6.11. Belgium

Belgium has no consensus on a GDM diagnosis, two methods are used at the same time. Additionally, there is a lack of accurate data on the GDM prevalence in this country [141]. Benhalima et al. compared the Carpenter and Coustan (C&C) criteria- old criteria and the IADSPG guidelines analyzing data from 6727 pregnant women. Using the old guidelines the GDM prevalence was 3.3% and using the new it was 5.7% [142]. According to study performed by Costa et al., the authors also presented the increase of GDM incidence between two- and one-step screening from 3.4% to 16.28% [143]. The one-step criteria were endorsed by Gropuement des Gynécologues Obstétriciens de Langue Française de Belgique (GGOLFB) [143]. Another study showed the GDM prevalence in Belgium using different criteria: IADPSG, NICE from 2015, Irish from 2010, French from 2010, and Dutch from 2010. The results presented 12.5%, 6.5%, 7.9%, 8.0%, and 8.9% GDM incidence, respectively [144].

### 6.12. Netherlands

The GDM prevalence in the Netherlands is estimated at approximately 5% using the WHO 1999 criteria [145]. Konnig et al. compared two screening tests to diagnose GDM: WHO 1999 and AIDPSG/WHO 2013 in women with a high risk for GDM development. There was a 45% increase in the number of diagnosed cases (32% in the AIDPSG/WHO 2013 group, and 22% in the WHO 1999 group) [146]. There is still a debate, over which criteria are better [147]. Other researchers from the Netherlands using the IADPSG/WHO 2013 criteria diagnosed GDM in 8.2% of patients and in 13.2% of GDM high-risk patients [148]. In a single center cohort study from Utrecht de Wit et al. confirmed an increase in GDM prevalence using the IADPSG/WHO 2013 criteria compared to the WHO 1999 strategy, at 32.4% vs. 19.3% [149]. Rademaker et al. wanted to show that the GDM incidence is higher among non Dutch-origin citizens of the Netherlands (e.g., Surinamese and Sub-Saharan African) pregnant women in comparison to Dutch pregnant women [150].

### 6.13. United Kingdom

In the United Kingdom the 75 g 2 h OGTT is used for pregnant women with risk factors for GDM, not at universal screening. The recommended criteria were prepared by NICE in 2015 and accepted by the Royal College of Obstetricians and Gynecologist (RCOG). According to them, a patient who has had GDM during a previous pregnancy, or is obese, or delivered a baby weighted at ≥4.5 kg, or one of the siblings has diabetes, or if the family origin is of South Asian, Chinese, African-Caribbean or Middle Eastern is offered early self-monitoring of blood glucose or a 75 g 2-h OGTT as soon as possible after a confirmation of pregnancy and then another OGTT between 24 and 28 weeks of gestation, if the results of the first OGTT are normal. GDM is diagnosed if the woman has a fasting plasma glucose level ≥ 5.6 mmol/L (≥100 mg/dL) or a 2-h plasma glucose level ≥ 7.8 mmol/L (≥140 mg/dL) [151,152]. The GDM prevalence is estimated at approximately 5% [153].

Garcia et al. analyzed data from Luton between 2008 and 2013 to assess the GDM prevalence among White British, Indian, Bangladeshi, and Pakistani pregnant women, which was the highest in Bangladeshi (2.1%) and the lowest in White British (0.4%). They used the NICE 2015 criteria retrospectively [154]. Plant et al. presented results from a study performed in Sandwell, West Midlands, a part of England where obesity, and physical inactivity levels are higher than the average. GDM prevalence was diagnosed in 6.8% cases using the NICE 2015 guidelines [155]. A similar result—at 6.77% was estimated by Martine-Edith et al. in Born in Bradford (BiB) cohort study using modified WHO 1999 criteria, according to local recommendations at the time of the study (2007–2010): fasting glucose level ≥ 6.1 mmol/L (≥110 mg/dL) and 2 h post-load glucose level ≥ 7.8 mmol/L (≥140 mg/dL) [156].

Data from Scotland, collected between 2010 and 2012, presented that for this part of the United Kingdon, the AIDPSG/WHO 2013 guidelines to diagnose GDM are recommended, but using only fasting and 2 h blood glucose levels [157]. In addition SIGN recommends the same criteria for diagnosing GDM but in high-risk patients [158]. Additionally, these criteria were not accepted by all obstetric units in Scotland, and universal screening was performed only in 20% of them. After changing the criteria, the GDM prevalence increased from 1.28% in 2010 to 2.54% in 2012 [157]. Collier et al. analyzed data from the Scottish Morbidity Record 02 between 1981 and 2012, and they showed 9-fold increase in the GDM prevalence during this time, with a result of 1.9% in 2012. In addition, they confirmed that BMI, maternal age, social deprivation, and multiparity were positively correlated with GDM [159].

### 6.14. Ireland

Just after the IADPSG released new GDM criteria, Ireland accepted them. According to the Atlantic Diabetes in Pregnancy (DIP), GDM was diagnosed in 12.4% cases. Additionally, there was a confirmation of the mothers age and obesity as risk factors for GDM [160]. Similar outcomes were assessed by Bogdanet et al., in their study GDM prevalence was 11.4% [161]. In another Atlantic DIP study, the higher prevalence of GDM in the lowest socioeconomic group was found. The increase was approximately 8.6% [162]. McMahon et al. analyzed data from an Irish hospital between 2008 and 2017. They presented a five-fold increase in the GDM incidence from 3.1% in 2008 to 14.8% in 2017, which is closely connected with changing guidelines for GDM diagnosis [163].

### 6.15. Hungary

Hungary accepted the AIDPSG guidelines for diagnosing GDM. Using this criteria Kun et al. estimated the GDM prevalence in Hungary at 16.6% retrospectively [164]. Data collected between 2009 and 2017 and presented also by Kun et al. showed an increase in the GDM incidence from 12.4% in 2009 to 18.5% in 2017 [165].

### 6.16. Romania

Romania accepted the IADPSG guidelines for diagnosing GDM, but the OGTT test is performed only if risk factors for GDM are confirmed [166]. Chelu et al. presented data from 2017 to 2021, in which the average GDM prevalence was estimated at 5.78% with the lowest result of 2.77% in 2017 and the highest outcome in 2021 at 8.48% [167]. Preda et al. described which medical situations were risk factors for GDM. According to this study, hypertension, gestational hypertension, history of fetal macrosomia, GDM in previous pregnancies, maternal age, and weight gain during pregnancy were significantly correlated with GDM [166].

### 6.17. Iceland

The GDM prevalence in Iceland is estimated between 15.5 and 19% using the IADPSG criteria [168,169]. A similar result in the GDM incidence was presented by Tryggvadottir et al., at 14.9% [170]. Another study from Iceland showed how maternal dietary patterns during pregnancy influenced the diagnosis GDM. In patient with a prudent dietary pattern and correct BMI, GDM appeared in 2.3% cases and among 18.3% of overweight/obese patients [171].

### 6.18. Denmark

A national study in Denmark on the GDM prevalence was performed between 2004 and 2012 among 566,083 patients. The results showed an increase in GDM incidence from 1.7% in 2004 to 2.9% in 2012. During this period of time to diagnose GDM was used the OGGT with 75 g glucose as a risk-factor screening. The criteria were a blood glucose level ≥ 9.0 mmol/L (≥162 mg/dL) in the capillary full blood/venous plasma or ≥10.0 mmol/L (≥180 mg/dL) in capillary plasma 2 h post-load [172]. Another study from Denmark conducted in years 2004–2017 estimated GDM prevalence on 2.5% [173]. Till 2013 OGTT was performed at 27–30 weeks of gestation, nowadays at 24–28 weeks of gestation. Nielsen et al. wanted to present situation, where GDM was diagnosed in 2.7% cases, but 23% from these cases were migrants [174].

### 6.19. Norway

There is a lack of systematically synthesized and integrated data on the GDM incidence in Norway [175]. For the last 20 years, three ways to diagnose GDM (i.e., the WHO 1999; the AIDPSG/WHO 2013 and the Norwegian guidelines) were used. The last criteria are defined as a fasting blood glucose level between 5.3 and 6.9 mmol/L (95 and 124 mg/dL) and/or a 2 h level of blood glucose between 9.0 and 11.0 mmol/L (162 and 198 mg/dL) after an OGTT [176]. Using this three ways, Rai et al. assessed GDM prevalence at 10.3% (WHO 1999); 10.7% (Norwegian); and 16.9% (IADPSG/WHO2013) [176]. Other researchers confirmed that GDM was more frequently diagnosed in immigrants from South Asia than in women of Norwegian origin [177].

Founger et al. showed that GDM was much more frequently diagnosed in pregnant women with polycystic ovary syndrome (PCOS) than in the general population. This depended on which criteria were used, as the GDM prevalence was 27.2% using the Norwegian criteria, 28.3% using the WHO 1999 criteria, and 41.2% using the AIDPSG/WHO 2013 criteria [178].

### 6.20. Sweden

A population- based cohort study from Sweden, performed between 1998 and 2012, assessed GDM prevalence for 1%. GDM was diagnosed in several ways, based on OGTT, during this time [179]. In another study, Nilsson et al. estimated the incidence of GDM in south Sweden at 2.2% for the years 2012–2013. They used a 75 g OGTT with a 2 h cut-off value of 10 mmol/L (180 mg/dL) to diagnose GDM [180]. In 2015 the Swedish National Board of Health recommended the IADPSG/WHO 2013 criteria for a GDM diagnosis [181].

### 6.21. Finland

According to the Finnish Current Care Guidelines for GDM, since 2008 a 75 g OGTT has been performed among all pregnant women without a very low risk of GDM. The diagnostic thresholds are: a fasting plasma glucose ≥ 5.3 mmol/L (≥92 mg/dL), 1 h glucose ≥ 10.0 mmol/L (≥180 mg/dL), and a 2 h glucose ≥ 8.6 mmol/L (≥155 mg/dL). The prevalence of GDM in primiparous women in Finland was assessed at 16.5% [182]. Koivunen et al. compared the IADPSG and NICE to diagnose GDM among 4033 pregnant women, and the results were 31.0% and 13.1% respectively [183]. In another study, Rönö et al. estimated a GDM prevalence of 13.0% in first the pregnancy, and 17.6% in the second pregnancy. GDM recurrence rate was 62.8% [184]. Ellenberg et al., analyzing data from the Hospital Discharge Register and the Finnish Medical Birth Register in the years 2006–2008 and 2010–2012 found an increase in GDM incidence from 7.2% to 11.2% [185].

### 6.22. Slovenia

In June 2011, Slovenia accepted the IADPSG guidelines and started universal screening for GDM. Before this date a two-step test that used C&C criteria for pregnant women with risk factors for GDM were performed [186]. Lucovnik et al. analyzed the data before and after changing the GDM diagnostic criteria from 276 210 deliveries. An increase in the GDM prevalence from 2.6% to 9.7% of cases was observed [186].

### 6.23. Croatia

Croatia used the WHO 1999 criteria before changing to the IADPSG guidelines. The Croatian Perinatology Society strongly advocated to make this change. The Croatian Chamber of Medical Biochemists appointed a working group to correctly perform the new procedures [187]. Djelmis et al. compared the estimated prevalence of GDM using different criteria: the IADPSG and NICE. In this study 4646 pregnant women took part. In 23.1% of cases, GDM was diagnosed using an OGTT test performed between 24 and 32 weeks of gestation according to the IADPSG guidelines. The NICE criteria met 17.8% of patients [188].

### 6.24. Serbia

In Serbia the ADA or IADPSG criteria for GDM diagnosis are used [189,190,191]. Earlier, the 100 g OGTT was used in high-risk patients [189]. Lackovic et al. assessed the incidence of GDM at 24.6%. In addition, they found positive correlations between GDM and BMI at delivery, gestational weight gain (GWG), pre-pregnancy BMI, positive family history for cardiovascular disease, LGA, mode of delivery, congenital thrombophilia, and hypertension in pregnancy [190]. Perovic estimated the GDM prevalence in high-risk patients for 25.7% [189]. A lower GDM incidence (19.5%) was presented by Milovanovic et al. in their study assessing usefulness of thyroid screening in GDM prediction [191].

### 6.25. Macedonia/Northern Macedonia

According to data obtained from Macedonia/Northern Macedonia the AIDPSG guidelines are used to diagnose GDM. An OGTT test with 75 g of glucose is performed between 24 and 28 weeks of gestation [192]. The only article, that presented the GDM prevalence in Macedonia/Northern Macedonia assessed the incidence at 66.1% [193].

### 6.26. Bosnia and Herzegovina

The latest data on the prevalence of GDM from Bosnia and Herzegovina are dated for the years 2010–2011. A total of 285 pregnant women with singleton pregnancies participated in the study. They underwent a 75 g OGTT between 22 and 32 weeks of gestation. The incidence of GDM was assessed at 10.9% according to the WHO 1999 criteria. Prenatal cigarette smoking, C-section rate, GDM in a previous pregnancy, and neonatal hypoglycemia were significantly more frequent in the GDM group compared to the controls [194].

### 6.27. Albania

The only study from Albania describing GDM prevalence and screening was performed between 2005 and 2012. GDM was diagnosed if the fasting plasma glucose was ≥120 mg/dL or the postprandial glucose level was ≥180 mg/dL. The prevalence of GDM was estimated at 2.8% [195].

### 6.28. Estonia

The actual criteria for a GDM diagnosis in Estonia were established in 2011. That year, the Estonian Gynecologists’ Society approved new guidelines based on the IADPSG recommendations. Estonians perform a 75 g OGTT test in pregnant women with a risk of GDM. Two articles presented the prevalence of GDM in Estonia [196,197]. Kirss et al. conducted their research in 2012, and the GDM incidence was 6.0% [196]. Another study showed results from between November 2013 and March 2015, the GDM prevalence was estimated at 28.1% [197].

### 6.29. Lithuania

According to researchers from Vilnius, the GDM prevalence in Lithuania increased 6.7-fold from 2001 to 2014 (2.7%). They did not write which criteria were used to diagnose GDM, but underlined that GDM screening became universal in Lithuania after decision made by the Lithuanian Ministry of Health [198]. Ramonienė et al. compared the diagnostic criteria for GDM proposed by the WHO 1999 and the IADPSG. The participants underwent a 75g OGTT between 24 and 28 weeks of gestation. All glycemia values were significantly higher in obese women than in the normal weight women group. According to the WHO 1999 criteria, GDM was diagnosed in 6.9% of obese patients vs. 2.9% in the controls (*p* = 0.195, OR 2.43 (95% CI (0.61–9.68)). Using the IADPSG criteria, GDM was diagnosed in 41.2% cases of obese pregnant women vs. 9.8% in the control group (*p* = 0.0001, OR 6.44 (95% CI (3.00–13.81)) [199].

### 6.30. Cyprus

Soytac Inancli et al. performed a study between 2013 and 2014 using 100 g OGTT, according to the National Diabetes Data Group (NDDG), between 24 and 28 weeks of gestation. The GDM prevalence was 19.6% among Turkish Cypriot [200].

### 6.31. Malta

In Malta screening is based on the IADPSG guidelines, but only high-risk patients are qualified for an OGTT test between 24 and 28 weeks of gestation [201]. Xuereb et al. using the IADPSG and WHO 2006 criteria estimated GDM prevalence in 21.2% cases with universal screening [202]. Cuschieri et al. decided to perform universal screening in their study, and GDM was diagnosed in 136 cases (33.9%) using the OGTT according to the IADPSG criteria [201].

### 6.32. Bulgaria

According to Bojadzhieva et al., the GDM prevalence in 2010 was estimated at 11.3% using the ADA criteria for GDM diagnosis [203]. Later Bulgaria accepted the IADPSG criteria to diagnose GDM. A study performed by Borrisov et al. showed that the incidence of GDM using the latest guidelines was assessed at 13.2%. Additionally, they found positive correlations between obesity, maternal age, GDM in previous pregnancy, family history of diabetes previous GDM, and high blood sugar before pregnancy and GDM [204].

### 6.33. Slovakia, Ukraine, Belarus, Latvia, San Marino, Liechtenstein, Monaco, Luxembourg, Moldova, Andorra, Georgia, Armenia, and Azerbaijan

Studies concerning GDM prevalence and screening in Slovakia, Ukraine, Belarus, Latvia, San Marino, Liechtenstein, Monaco, Luxembourg, Moldova, Andorra, Georgia, Armenia and Azerbaijan were not found.

## 7. Asia

The following areas of Asia were distinguished: North Asia, Central Asia, East Asia, South Asia and West Asia. It is estimated that GDM in Asian countries ranges from 1.2 to 49.5% (Figure 2). The available reports on the incidence of GDM provide very diverse data, which may be related to the divergence in the methods of estimating the prevalence and different diagnostic criteria in individual countries [205].

### 7.1. Central Asia

Central Asia is a region, that includes Kyrgyzstan, Kazakhstan, Uzbekistan, Tajikistan and Turkmenistan.

#### 7.1.1. Turkmenistan

In Turkmenistan, between March 2008 and March 2011, Parhofer et al. conducted a screening program to assess the incidence of GDM. A total of 1738 pregnant women took part in the test which relied on a 50 g GCT. All of the participants were at 34 weeks of gestation. If the glucose reference value was ≥7.8 mmol/L (≥140 mg/dL), an OGTT with 75 g oral glucose should be performed. The disease was recognized when ≥1 glucose values were abnormal; ≥5.0 mmol/L (≥90 mg/dL) at 0 min, ≥10.0 mmol/L (≥180 mg/dL) at 60 min and ≥8.0 mmol/L (≥144 mg/dL) at 120 min. A positive screening test was confirmed in 22.7% of the patients. A total of 70 % of them underwent an OGTT test, and 39.5% of the women were diagnosed with GDM. The overall prevalence was assessed at 6.3% [206].

#### 7.1.2. Tajikistan

In Tajikistan the WHO 2013 guidelines are used to detect GDM. Between September 2015 and November 2017 Pirmatova et al. conducted a cohort-study of 2438 pregnant women. The patients were between 24–28 weeks of gestation. Referring to the WHO 2013 criteria, the scientists found a very high rate of GDM in Tajikistan which was approximately 32.4%. The patients with GDM were older, and had a higher BMI [207].

#### 7.1.3. Kyrgyzstan

Unfortunately, today, no data exist that could assess the prevalence of GDM. It is speculated that the occurrence of GDM may be estimated to a similar degree as that in Turkmenistan [208].

#### 7.1.4. Uzbekistan

Between 2017 and 2020 the international project “Strategy for the Prevention and Monitoring of GDM in Uzbekistan was performed. According to the recommendations of the ADA 2015, an OGTT test with 75 g of glucose was conducted. After 60 min, a significantly high level of glucose was observed in 53.7% of the participants. A total prevalence was assessed at 10.5% [209].

#### 7.1.5. Kazakhstan

At the moment, there are no statistically reliable data regarding the prevalence, guidelines and risk factors for GDM in women living in Kazakhstan [210].

### 7.2. North Asia

#### Siberia

At present, there are no scientific studies that could describe the incidence of GDM in the Siberian population [211].

### 7.3. South Asia

The South Asian Federation of Endocrine Societies (SAFES) includes The Endocrine Society of Bangladesh, Endocrine Society of India, Diabetes and Endocrine Association of Nepal, Pakistan Endocrine Society, and Endocrine Society of Sri Lanka. At the turn of 2015 new recommendations for the diagnosis and treatment of GDM were issued and the disease itself was made a health priority [212].

#### 7.3.1. Nepal

During the period from July 2009 to June 2010 in three districts of Nepal Thapa et al. conducted a study with the main aim of assessing the prevalence of GDM. According to the WHO criteria approximately 2.5% had GDM and while using the IADPSG criteria 6.6% had positive results [213]. In 2022 Pashupati et al. analyzed 20,865 cases and estimated the total incidence of GDM at 6.56% using the IADPSG criteria, 4.81% using the (WHO) criteria and 4.71% using the Diabetes in Pregnancy Study Group of India (DIPSI) guidelines [214].

#### 7.3.2. Bangladesh

In Bangladesh, in 2014 Jesmin et al. performed a study in which 3447 pregnant women took part. The WHO and ADA criteria were used to assess the total prevalence of GDM. The results were 9.7% according to the WHO 1999 guidelines and 12.9% according to the ADA criteria [215]. Between 2017 and 2018 an attempt to evaluate the national incidence of GDM was made. The overall prevalence of GDM in Bangladesh was 35% (95/272) [216].

#### 7.3.3. Sri Lanka

In Sri Lanka two Medical Offices of Health (MOH) conducted a cross-sectional study between January 2014 and March 2015. A GDM diagnosis was based on a fasting 75 g OGTT according to the WHO 1999. Positive results were noted in 13.9% of the participants [217]. After adopting the IADPSG criteria to diagnose GDM, the total prevalence of the disease was 31.2% [218].

#### 7.3.4. Pakistan

Between January and August 2017 Wali et al. conducted a study with the main goal of assessing the prevalence of GDM and 21.8% of pregnant women were diagnosed with GDM [219].

#### 7.3.5. India

The year 2014 was a breakthrough for India when the Government of India (GoI) mandated universal GDM screening for all pregnant women as part of essential obstetric care within the Reproductive and Child Health (RCH) program. At that time a pooled GDM prevalence was assessed at 8.9%. In 2018 subset meta-analyses showed that the IADPSG diagnostic criteria found significantly more GDM cases with a total incidence of 19.19% in comparison to the WHO 1999 criteria at 10.13%. The DIPSI criteria estimated the total prevalence of GDM at 7.37% [220].

### 7.4. East Asia

#### 7.4.1. South Korea

Over the period 2007–2011 data obtained from the Health Insurance Review and Assessment (HIRA) database were analyzed in terms of women’s age, and the number of pregnancies in particular years. The results showed that the incidence of GDM during that period was 7.5% in 2009–2011, 5.7% in 2009, 7.8% in 2010, and 9.5% in 2011 [221]. Between 2012 and 2016 Jung et al. performed a cross-sectional study on the basis of which the GDM occurrence was assessed at 11.1% [222]. Cha et al. performed a study in 2022 which main goal was to simplify the diagnosis of GDM. In South Korea the two-step approach according to the ACOG (a universal 50 g OGTT followed by diagnostic 100 g 3-h OGTT) is used to diagnose GDM. The medical histories of 1441 pregnant women were analyzed. If a glucose value ≥ 7.8 mmol/L (≥140 mg/dL) obtained via the screening with a 50 g glucose load was considered to be positive, a diagnostic 100 g OGTT test had to be performed as the next step. The C&C criteria for a GDM diagnosis were used. A total of 93 out of 423 (22%) in this group were diagnosed with GDM according to the C&C criteria. The upper cutoff for a GDM diagnosis in the 50 g OGTT was >12.3 mmol/L (>222 mg/dL), and the lower was <7.3 mmol/L (<131 mg/dL). Previous studies using upper cut-offs in the 50 g OGTT suggested 185, 220, 228, or 230 mg/dL as the upper cutoff for the omission of the 100 g OGTT. The scientists proposed a 12.3 mmol/L (222 mg/dL) level in the 50 g OGTT [223].

#### 7.4.2. Japan

In 2010 the Japan Society of Obstetrics and Gynecology (JSOG) changed the diagnostic guidelines for detecting GDM to the IADPSG criteria [224]. Six years later a diagnosis of GDM was determined using the criteria of the Japan Diabetes Society (JDS) and a total prevalence was assessed at 2.30% [225]. In 2018 two tests for detecting GDM in pregnant women were compared. The percentage of women screened using 5.3 mmol/L (95 mg/dL) as the cut-off value for random plasma glucose was significantly higher in comparison to those who were tested using 5.55 mmol/L (100 mg/dL) as the cut-off value for random plasma glucose (2.7% and 6.9%, *p* < 0.0001). Moreover, women who were screened for GDM using random plasma glucose and a 50 g GCT had a significantly higher incidence of GDM (6.6% vs. 8.9%, *p* < 0.0001) [226].

#### 7.4.3. China

In China guidelines such as the IADPSG/China, WHO 2013, ADA 2012, ADA 2014, C-C ACOG and WHO 1999 are used to diagnose GDM. In 2019 He et al. performed a study that evaluated how different GDM diagnostic criteria influenced the national prevalence of GDM. They used data collected from women undergoing a 2-h, 75 g OGGT at 24–28 gestational weeks from January 2011 to December 2017 and they developed the results using different criteria (i.e., the 7th edition textbook criteria, NDDG 1979, WHO 1985, European Association for the Study of Diabetes 1996, Japan 2002, ADA 2011, and NICE Excellence 2015 criteria). The incidence of occurrence of GDM based on the ADA 2011 and NICE were 22.94% and 21.72%, over threefold higher than implementing the 7th edition textbook criteria. The incidence rates of GDM diagnosed with the NDDG1979 and WHO 1985 guidelines were significantly less than the 7th edition textbook criteria [227].

### 7.5. Southeast Asia

According to a report conducted in 2019 the overall prevalence of GDM was described for: Thailand and Singapore (24.7% vs. 23.5%), Malaysia (22.5%) and Vietnam (21.3%). Over the years 2010–2020 Malaysia used guidelines to detect GDM such as: theMalaysia MOE/NICE, WHO 1985, WHO 1994 and DIPSI or WHO 1999. Vietnam adopted the criteria of the IADPSG/China, WHO 2013, ADA 2012 and ADA 2014. The screening tests and recommendation followed by Singapore were: the IADPSG/China, WHO 2013, ADA 2012, ADA 2014, ADA 2007 and the 4th International Workshop-Conference on GDM [228]. In some regions of Southwest Asia HbA1c is used. Tests that can also be widely used are the one-step method comprising a 75 or 100 g OGTT and the two-step method with a 50 g OGT. In Thailand the most frequently used method was the one-step approach [229].

The 100 g, three-hour OGTT was believed to be the gold standard in Southeast Asia to detect GDM in pregnant women [229]. Based on the plasma glucose level two methods are distinguished—the C&C values and the NDDG values [228]. To confirm GDM two abnormal parameters are needed. In Thailand, an increase in the prevalence of GDM from 22.2% to 32.76% was observed when changing from the C&C criteria to the NDGG criteria. The researchers showed that the most valuable screening test in Thailand was the 100 g, two-hour OGTT [229]. In Vietnam GDM was diagnosed in 6.1% using the ADA criteria and in 20.3% using the IADPSG criteria [230]. The WHO diagnostic criteria were established in 2013. They include the 75-g, two-hour OGTT test. The new guidelines for detecting GDM lower the total prevalence of the disease in Singapore [231]. Many scientific studies emphasize the greater legitimacy of using a two-hour OGTT instead of a three-hour OGTT.

Recently there has still been a lack of consensus regarding the use of diagnostic criteria for GDM in Southeast Asia. Lowering the two-hour OGTT threshold values may detect more cases of GDM and enable the use of an appropriate treatment as soon as possible [228].

### 7.6. West Asia

In the western part of Asia, the one-step test to detect GDM is mostly used in countries such as Saudi Arabia, Quatar, Yeman, and the UAE. The two-step test is more popular in Oman, Bahrain and Turkey [2]. The use of different screening methods for GDM for each country is as follows: Yeman—WHO 1998 and ADA 2002 criteria; Quatar—WHO 2006, IADPSG/China, WHO 2013, ADA 2012, ADA 2014, and ADA 2004; Saudi Arabia—IADPSG/China, WHO 2013, ADA 2012, ADA 2014, C-C ACOG, ADA 2011, and ADA 2007; Iran—Kuwait self-report GDM, IADPSG/China, WHO 2013, ADA 2012, ADA 2014, C-C, and ACOG; Kuwait—Kuwait self-report GDM; Turkey—C-C ACOG; Turkey—NDDG, IADPSG/China, WHO 2013, ADA 2012, ADA 2014, C-C, and ACOG; Bahrain—NDDG, ADA 2007; Oman—Oman self-defined guidelines [2,232].

#### 7.6.1. Saudi Arabia

Saudi Arabia is on the top ten countries in the world with the highest prevalence of T2DM. It is estimated that the highest GDM prevalence is approximately 49.5%. The incidence of GDM was higher in the 31–35 age group. The reason for such a high score may be a genetic predisposition to insulin resistance compared to Caucasians. Within the same country, different diagnostic criteria were used to diagnose GDM. Furthermore, different cut-off values of 5.1 mmol/L 92 mg/dL or 5.3 mmol/L 95 mg/dL for the 75 g OGTT were used [2].

#### 7.6.2. Oman

In 2013 Oman introduced new criteria to detect GDM. A total of 613 Omani women took part in a study conducted by Subshi et al., which was based on the current diagnostic criteria, and the incidence of GDM was 48.5%. It dropped to 26.4% when applying the new WHO criteria [232]. In another study performed by Chitme, the glucose profile, family history, anthropometric profile, and age of first pregnancy were analyzed. Patients who had a fasting plasma glucose ≥ 5.6 mmol/L and/or two-hour-oral glucose tolerance test ≥ 7.8 mmol/L were considered to have GDM. Eleven percent of pregnant women developed GDM [233].

#### 7.6.3. Kuwait

In Kuwait the occurrence of GDM is commonly associated with poor maternal, fetal, and neonatal outcomes. In 2019 a cross-sectional study was conducted and 947 pregnant women took part in the screening. Of the 868 mothers with no prior history of diabetes mellitus, 109 reported were given a GDM diagnosis, resulting in 12.6% [234].

#### 7.6.4. Qatar

Out of a total of 17,020 live births in 2017, 5195 newborns were born to Qatari women. Of these, 1260 were born to women with GDM. The prevalence of GDM in the Qatari population in 2017 was 24.25%. The HbA1C% before delivery was significantly higher in women with GDM in comparison to healthy ones. A higher maternal age and obesity were significantly associated with an increased risk of GDM [235].

#### 7.6.5. Iran

In Iran during 2015–2016, 1010 pregnant women took part in a screening program. The risk of GDM was 10.1%. Due to the political situation no recent data concerning GDM prevalence exist [236].

#### 7.6.6. Iraq

The GDM prevalence in 2014 in Iraq was estimated on 7% [237]. In 2020 120 pregnant women took part in a study, in which approximately 13.3% GDM was detected [238].

#### 7.6.7. Bahrain

In 2012, GDM was assessed at 10.1%. Between 2002 and 2010, there was an increase in GDM detection from 7.2% to 12.5%. The main risk factors are weight and maternal age [239].

#### 7.6.8. Palestine

A prevalence of DM in Palestine was estimated at 20.8% in 2020. GDM is hard to assess [240].

#### 7.6.9. Jordan

In Jordan between 2015 and 2016, the GDM prevalence was assessed on 13.5%. Maternal age, parity, gravidity, maternal BMI and pre-pregnancy BMI were risk factors [241].

#### 7.6.10. Yeman

In 2019, it was assessed that approximately 3.9% of women had GDM. A family history of GDM, age > 30 years, history of PCOS, and previous GDM were risk factors. Due to the political situation no recent data concerning GDM prevalence exist [242].

#### 7.6.11. United Arab Emirates (UAE)

In the UEA different kind of criteria for diagnosis GDM exist. The prevalence of GDM in the UAE varies from 7.9 to 37.7%. The main risk factors for GDM development are parity, obesity, and glucose intolerance in young girls [242].

#### 7.6.12. Lebanon

No data exist concerning GDM prevalence [243].

#### 7.6.13. Syria

No data exist concerning GDM prevalence [243].

## 8. Conclusions

Analyzing data from all over the world (Appendix A) we wanted to show that a lot of work is needed to achieve a consensus in the diagnostics of GDM. New studies are crucial to finding a solution. We hope that they will confirm and persuade scientists to choose the best method to diagnose GDM.

Although today we detect GDM radically differently than in earlier centuries—the disease is still a serious challenge for medicine around the world. Looking at the history of diabetes, we are sure that more than one evolution in GDM diagnosis will occur, due to the development of medicine, appearance of modern technologies and the dynamic continuation of research.

## Figures and Tables

**Figure 1 ijerph-19-15804-f001:**
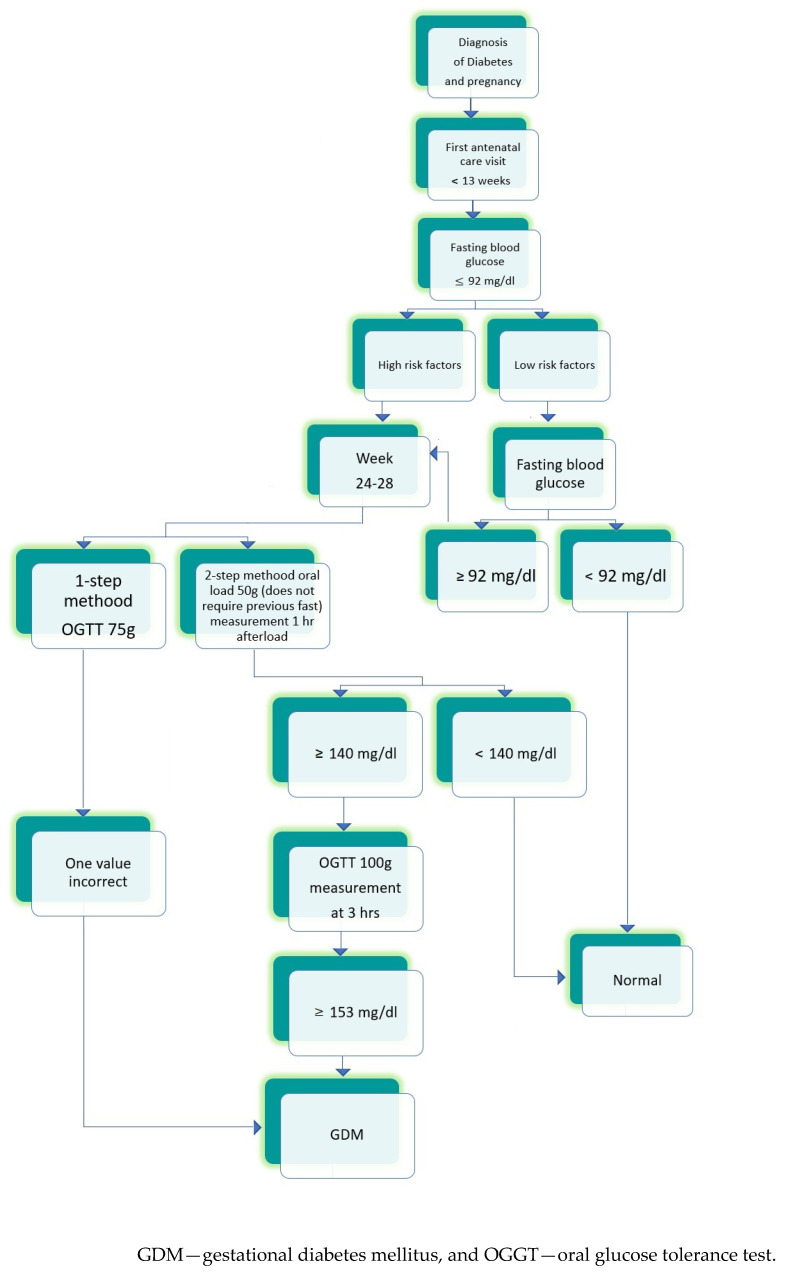
The scheme prepared according to “Guia de practica clinica Diagnostico y tratamiento de diabetes en el embarazo, CENETEC 2016” [53].

**Figure 2 ijerph-19-15804-f002:**
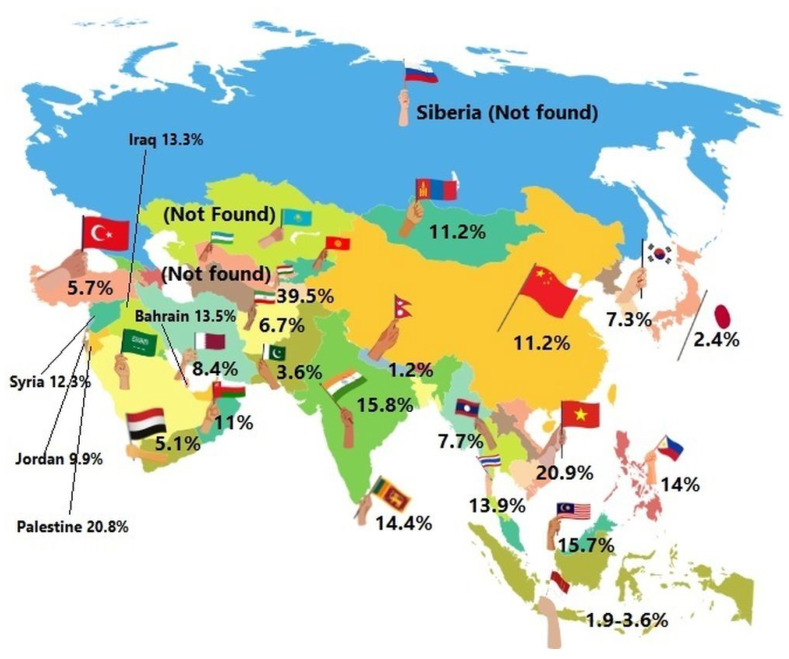
Gestational diabetes mellitus in Asia—the estimated prevalence is shown in percentages across the countries, 2022.

**Table 1 ijerph-19-15804-t001:** Criteria for gestational diabetes mellitus screening by selected societies [12].

Society	Test	Number of Abnormal Values Required for Diagnosis	Fasting Glucose (mg/dL)/(mmol/L)	1 h after Loading (mg/dL)/ (mmol/L)	2 h after Loading (mg/dL)/ (mmol/L)	3 h after Loading (mg/dL)/(mmol/L)
ACOG 2017/C-C	Two step 3 h 100 g	≥2	95/5.3	180/10.0	155/8.6	140/7.8
ACOG 2017/NDDG	Two step 3 h 100 g	≥2	105/5.8	190/10.5	165/9.2	145/8.1
ADA 2017 75 g	One step 2 h 75 g	≥2	95/5.3	180/10.0	155/8.6	-
ADA 2017 100 g	Two step 3 h 100 g	≥2	95/5.3	180/10.0	155/8.6	140/7.8
CDA 2013	One step 2 h 75 g	≥2	95/5.3	191/10.6	160/8.9	-
FIGO 2013/ WHO 2013/ IADSPG 2013	One step 2 h 75 g	≥1	92/5.1	180/10.0	153/8.5	-
NICE/RCOG 2015	One step 2 h 75 g	≥1	101/5.6	-	140/7.8	-
WHO 1999	Fasting OGTT with 75 g	1	-	-	≥140/≥7.8	-
DIPSI	Nonfasting OGTT with 75 g	1	-	-	≥140/≥7.8	-

ACOG—American College of Obstetricians and Gynecologists, ADA—American Diabetes Association, CDA—Canadian Diabetes Association, C-C—Carpenter and Coustan, FIGO—International Federation of Gynecology and Obstetrics, IADPSG—International Association of Diabetes Pregnancy Study Group, NICE—National Institute for Health and Care Excellence, RCOG—Royal College of Obstetricians and Gynecologists, NDDG—National Diabetes Data Group, WHO—World Health Organization, and DIPSI—Diabetes In Pregnancy Study Groups of India.

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
