# Peer review of "Evolution of Gestational Diabetes Mellitus across Continents in 21st Century"

_ijerph, 2022, doi:10.3390/ijerph192315804_

Round 1
Reviewer 1 Report
I carefully read what you wrote in the review sent for publication and I find it interesting, well documented and welcome in the current context. In the last decade, gestational diabetes experienced a recrudescence in terms of the number of discovered cases and the specialists in the field paid due attention to it. It is to be appreciated that apart from the statistical data presented for continents and countries, the diagnostic criteria as well as the updated therapeutic approaches are also presented.
It would be necessary to add some studies published in 2021 in some European countries and published in prestigious professional journals and to complete with these studies for the countries where less has been written.
Reviewer 2 Report
This is a well documented paper that lists all the inconsistent criteria to diagnose GDM for different countries due to ethnicity and geographical factors. The authors did a very thorough review for all countries. However, it is a little difficult to follow the minor differences across countries or continents. It would be helpful to have some tables to list out the differences for each country especially if a variety of standards exists, and also tables for horizontal comparison across continents.
Reviewer 3 Report
REVIEW of the study by Dluski DF, Ruszala M, Rudzinski G, Pożarowska K, Brzuszkiewicz K & Leszczynska-Gorzelak B: Gestational diabetes mellitus across continents in XXI century.
This is an interesting and comprehensive review that provides an overview of diagnostic criteria as well as an approach to Gestational Diabetes Mellitus (GDM) across several countries from all inhabited continents. The Authors’ managed to demonstrate differences in diagnostic algorithms in various health care systems as well as the lack of universally recognised diagnostic criteria, differences pertaining the role of 50 g glucose challenge test, etc.
The Authors also state (2nd & 3rd line of the main text) that “There is currently not enough research to show which way is the best at diagnosing GDM”. In view of that, in my opinion, the issue of screening for early, i.e. first trimester GDM should be at least mentioned in the first chapter as there are several papers on this subjects from countries on various continents, e.g. Sesmilo G, Prats P, Garcia S, et al. Acta Diabetol 2020; 57: 697-703, Kuehn J, Gebuerhr A, Wintour J et al., Aust N Z J Obstetr Gynaecol 2021; 61: 142-146, Tong JN, Wu LL, Chen YX et al. BMC Pregnancy Childbirth. 2022; 22: 540.doi: 10.1186/s12884-022-04874-x (role of fasting glucose), Valadan M, Bahramnezhad Z, Golshahi F, et al. BMC Pregnancy Childbirth 2022 22: 71, doi: 10.1186/s12884-021-04330-2, Kattini R, Hummelen R, Kelly L. J Obstetr Gynaecol Can 2020; 42: 1379-1384 (role of HbA1c), Corrado F, D’Anna R, Cannata ML et al, Diabetes Metab 2012; 38: 458-61 (role of fasting glucose & first trimester OGTT), including the data from the Authors’ own country (Lewandowski K, Gluchowska M, Garnysz K et al. Endokrynol Pol. 2022;73(1):1-7.doi: 10.5603/EP.a2021.0095. – OGTT and insulin resistance in 1st trimester). Furthermore, the issue of first trimester screening should be at least mentioned as Authors’ admit that selective 1st trimester screening is already recommended in some countries, such as the UK, where a patient who had GDM in previous pregnancy is offered “a 75 g 2-hour OGTT as soon as possible after confirmation of pregnancy, and then another one between 24 – 28 weeks of gestation…” (line 672-674).
Minor points:
line 258: The National Institutes (not Institute) of Health
Line 380: GDM can be diagnosed, when….
Reviewer 4 Report
Too many unnecessary details
Round 2
Reviewer 1 Report
As we pointed out in the previous review, the team that developed the article made a special effort to update the data on the epidemiology of gestational diabetes on the world map.
The new version of the review proves your team's desire to provide a picture as close as possible to the reality of the spread of gestational diabetes, so I appreciate the effort made to remove some passages that unnecessarily burden the article and to bring to the readers' attention new data related to diabetes gestational gathered from recently published articles.
Reviewer 4 Report
better, but still too big with unnessesary details